# Modelling Projected Changes in Soil Water Budget in Coastal Kenya under Different Long-Term Climate Change Scenarios

**Cornelius Okello [1,2,]***, **Nicolas Greggio [3]**, **Beatrice Maria Sole Giambastiani [3]**, **Nina Wambiji [4]**, **Julius Nzeve [1]** and **Marco Antonellini [3]**

1   Department of Environmental Science, Machakos University, Machakos P.O. Box 136-90100, Kenya; jnzeve@mksu.ac.ke
2   Institute of Climate Change and Adaptation, Chiromo Camus P.O. Box 30197-00100, Nairobi, Kenya
3   Department of Biological, Geological and Environmental Sciences, Alma Mater Studiorum-University of Bologna, Ravenna Campus, Via San Alberto 163, 48123 Ravenna, Italy; nicolas.greggio2@unibo.it (N.G.); beatrice.giambastiani@unibo.it (B.M.S.G.); m.antonellini@unibo.it (M.A.)
4   Kenya Marine and Fisheries Research Institute, Mombasa P.O. Box 81651-80100, Kenya; nwambiji@gmail.com
*   Correspondence: cbokello@gmail.com; Tel.: +254-733703940

**Abstract:** The possible impacts that climate change will have on soil water budget and specifically on deep percolation, runoff and soil water content have been investigated using HYDRUS, a methodology based on numerical modelling simulations of vertical water movement in a homogenous soil column on a flat surface. This study was carried out on four typical soil types occurring on the Kenyan coast and the adjacent hinterlands of up to an elevation of 200 m above sea level (m a.s.l.) covered by five weather stations (two dry and three wet stations). Results show that deep percolation and runoff are expected to be higher in 2100 for both Relative Concentration Pathways (RCPs) 2.6 and 8.5 scenarios than they were for the reference period (1986–2005). The average deep percolation is expected to increase by 14% for RCP 2.6 and 10% for the RCP 8.5, while the average runoff is expected to increase by 188% and 284% for the same scenarios. Soil water content is expected to either increase marginally or reduce depend in the same scenarios. The average soil water content is also expected to increase by 1% in the RCP 2.6 scenario and to decrease by 2% in the RCP 8.5 scenario. Increase in deep percolation through clay soil is expected to be the largest (29% in both scenarios), while sandy and sandy clay soil are expected to be the least influenced with an average increase of only 2%. Climate change is expected to impact runoff mostly in sandy soils, whereas the least affected would be clay loam soils. These results further support the assertion that the change in climate is expected to impact the recharge of aquifers by triggering an increase in infiltration under both scenarios.

**Keywords:** soil water storage; recharge; climate change; soil; HYDRUS

---

## 1. Introduction

Groundwater resources are critical for human activities such as domestic consumption, agricultural uses and industrial processes. Of these resources, the most precious groundwater is found in the soil (soil water) for use of the crops and the vegetation as well as in deep aquifers that usually contain less contaminants [1]. The unsaturated zone in the soil profile (referred to as the vadose zone) is an integral component of the hydrological cycle that directly influences processes such as infiltration, surface runoff, evapotranspiration, interflow, residence time of aquifer recharge and water table fluctuations thus providing a complex system for the simulation of water movement into aquifers [2,3]. This makes

soil moisture in the vadose zone a key variable in hydrology and climate change studies as it is a store of water in the hydrologic cycle, making it central to the catchment scale water balance [4]. Therefore, the understanding of the water storage and flow through soils and the vadose zone is one of the essential hydrological processes with great relevance for the appropriate use, management and protection of water resources [5,6].

Groundwater recharge (herein referred to as deep percolation) has long been a focus for agriculture and water research [7]. In order to characterise deep percolation, the physical processes including soil controls such as soil chemistry, thickness, layering, vegetation cover, tillage, roughness, topography, temperature and rainfall intensity must be considered [8]. Soil properties, in particular, have been found to dramatically affect the direction and rate of water flow [6]. Once these specific soil characteristics have been described, analytical solutions for local infiltration can be applied [2]. Acquisition of soil moisture data and estimation of deep percolation currently relies on three approaches. First, it can be done through *in situ* (generally point-scale) measurements [9–11]. A second approach involves remote sensing observations such as those using Gravity Recovery and Climate Experiment (GRACE) satellite mission datasets [12–14]. Finally, hydrological modelling that range from empirical to physically based models can be used [4,5].

A large number of modelling approaches have been developed to simulate the temporal dynamic of soil moisture and evaluate deep percolation for use in a wide range of applications e.g., research, management and risk assessment of subsurface systems [15,16]. Many of these models are based on a conceptual representation of the system, which inherently involves a number of important assumptions [6]. Some of the models used are approximations (numerical models) [17–19], while others are analytical solutions [16,20,21]. These models cannot, however, provide the profile of water content without large simplifications of the soil description. As more complicated models are being developed, the accuracy of numerical simulations largely depends upon the accuracy with which various model parameters can be estimated.

Models for the unsaturated zone are often based on the numerical solutions of the Richards equation [22], which requires knowledge of the unsaturated soil hydraulic functions, i.e., the soil water retention curve, $\theta(h)$, describing the relationship between the water content, $\theta$, the pressure head, $h$ and the unsaturated hydraulic conductivity function, $K(h)$. Accurate measurement of these hydraulic properties is extremely difficult because of their nonlinear nature, especially $K(h)$, and the extreme heterogeneity of the subsurface environment [23]. In developing countries such as Kenya, these kinds of data are rare or completely missing due to a variety of reasons. For example, little has been done to assess and gather data on groundwater resources in this study area (coastal Kenya). Furthermore, according to population statistics [24], the study area also has a higher population with its majority living within 100 km of the Indian Ocean. This is important because the effects of anthropogenic activities will play a major role in the deep percolation processes for all types of soil: the effects of agriculture can be experienced through tillage of land that changes the infiltration and runoff characteristics of the land surface, thus affecting recharge to groundwater, delivery of water and sediment to surface-water bodies and evapotranspiration [25]. Deforestation to create space for agriculture and human settlements also tends to decrease evapotranspiration, increase storm runoff and soil erosion and decrease infiltration to aquifers and base flow of streams, especially as urbanization results in more paved (and therefore impervious) areas [26–28].

Methods for making relatively fast and reliable measurements of the unsaturated soil-hydraulic properties are sorely needed [29]. The Richards equation is often chosen to physically represent the fluxes into the vadose zone at the local and regional scale [30–32]. It establishes the liquid mass conservation through soils in gravity/pressure-driven flows. Although it is based on mass conservation, it commonly includes several significant approximations. Its simplest 1D version does not obviously consider the flow in the horizontal directions, and does not include either source or sink terms [6]. Richards' models rely on the Darcy–Buckingham constitutive equation for water movement in unsaturated soils, which may fail for swelling soils or preferential flow of water through large pores

in contact with free water. Despite all these limitations, the Richards model is amply recognised as the most appropriate framework by some infiltration studies, because it accurately describes such a process in many practical situations. In fact, it has been frequently used as reference against which to validate more empirical or simplified approaches [30–32].

Globally, future groundwater recharge will depend heavily on future climate conditions [33]. This is because the hydrological cycle is substantially influenced by climate change [34–38]. It is, therefore, of utmost importance to analyse the impact of climate change on hydrology at a regional or local scale in order to understand potential future changes of water resources availability and provide support for regional water management [39]. In hydrology, climate change is resolved in terms of precipitation, surface-air temperature, evapotranspiration, sediment transport, groundwater levels, water quality and runoff changes at the relevant spatial scales [40]. When modelling is used in hydrological studies, it is useful to quantify the climate components of a water budget, because water budgets are driven principally by precipitation (P) and evapotranspiration (ET) [2]. The changing climate will influence these two, directly impacting soil moisture and deep percolation.

Representative Concentration Pathways (RCPs) are climate change scenarios developed as a basis for assessment of possible climate impacts and mitigation options with associated costs as well as to provide plausible descriptions of how the future may evolve with respect to a range of variables including socio-economic, technological, energy and land use change, emissions of greenhouse gases and air pollutants [41]. They are classified as follows: RCP 8.5 which is considered a high emission, highly energy-intensive scenario as a result of high population growth and a lower rate of technology development [42]. It was based on a revised version of the Special Report on Emissions Scenarios (SRES) A2 scenario [43] where the storyline emphasises high population growth and lower incomes in developing countries. The RCP 2.6 is the lowest emission and radiative forcing scenario representative of mitigation measures aiming to limit the increase of global mean temperature to 2 °C [44]. It is important to understand the movement of water in the soil through the vadose zone and into the underlying aquifers under these two extreme climate change scenarios.

Impacts of climate change on hydrological systems have been widely investigated by a number of studies conducted worldwide [40,45–48], and specifically in China [34,39,49–51], the Chalk aquifer in the UK [52], in Ontario, Canada [53], in Chile [54], as well as in the Hanko aquifer in Finland [55]. The HYDRUS-1D model, has been employed successfully by several authors around the world punctuating its applicability on a global scale, especially when considering impacts of climate change [56,57] and soil types [4,58,59]—this supports the choice to use HYDRUS-1D for this study.

The main objective of this study was to model expected impacts that changes in precipitation and temperature brought about by climate change may have on soil water budget (water content and groundwater recharge). This was based on the analysis of actual and future (modelled) regional climate data, projected changes in climate, statistical weather realizations and the solution of Richards equation as implemented in the HYDRUS-1D model. Several studies have already carried out in similar Arid and Semi-Arid Lands (ASALs)/dry regions using the HYDRUS-1D model similar to large parts of the study area [60–62]. Considerations were stressed on the three soil types covering the largest area along the Kenyan coast and the adjacent hinterlands as well as the sandy soils of the Lamu sand dunes. While a number of studies have been done in Kenya on both infiltration and deep percolation [63–66], they have mainly focused on inland areas. Studies along the Kenyan coast still remain scarce. It is important to note that this is a pilot study in an understudied area with limited data with the aim to sparking further research in the area. Therefore, factors such as vegetation and land use were not considered in the study, even as we acknowledge that they are relevant factors. Soil type and changes in average annual precipitation and temperature were the main factors considered.

*Study Area*

The study area (Figure 1) is located in Kenya and lies between LONG 1°61′ N and LONG 4°68′ S and between LAT 39°00′ E and LAT 41°59′ E. It covers the coastal plains and the adjacent hinterlands

that extend up to an elevation of 200 m above sea level (m a.s.l.) covering a total of approximately $2.3 \times 10^6$ ha. The most common soil type is clay, which covers about 50% of the total area. Other major soil types are clay to sandy clay loam (13.7%), clay loam to loam (7.7%), sandy loam to sandy clay loam (6.5%) and clay loam (5.6%). For purposes of this study, the study area was assumed to be flat with an average slope of 5%.

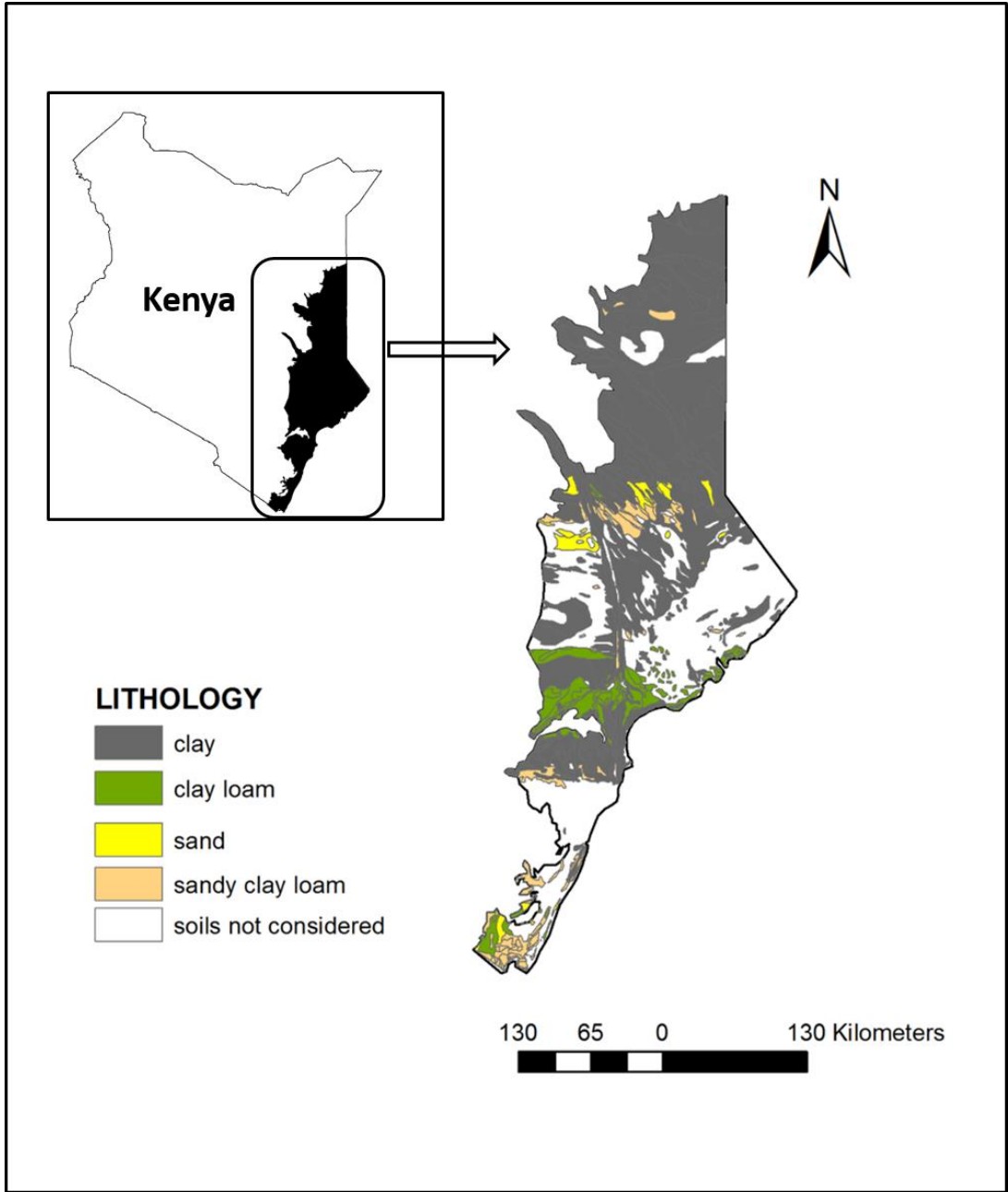

**Figure 1.** Map of the study area highlighting three most prevalent soil types as well as sand covering the dunes overlying the Shela aquifer on Lamu Island, Kenya.

The study area is covered by five weather stations monitored by the Kenya Meteorological Department (KMD) and their locations are reported in Figure 2. The precipitation for all five weather stations ranges from 8.1 to 102.2 cm/year, most of which fall between the months of March and May (long rain season) and September to November (short rain season). The annual average temperature is high, between 26 and 29 °C, and the potential evaporation is approximately 145–287 cm/year.

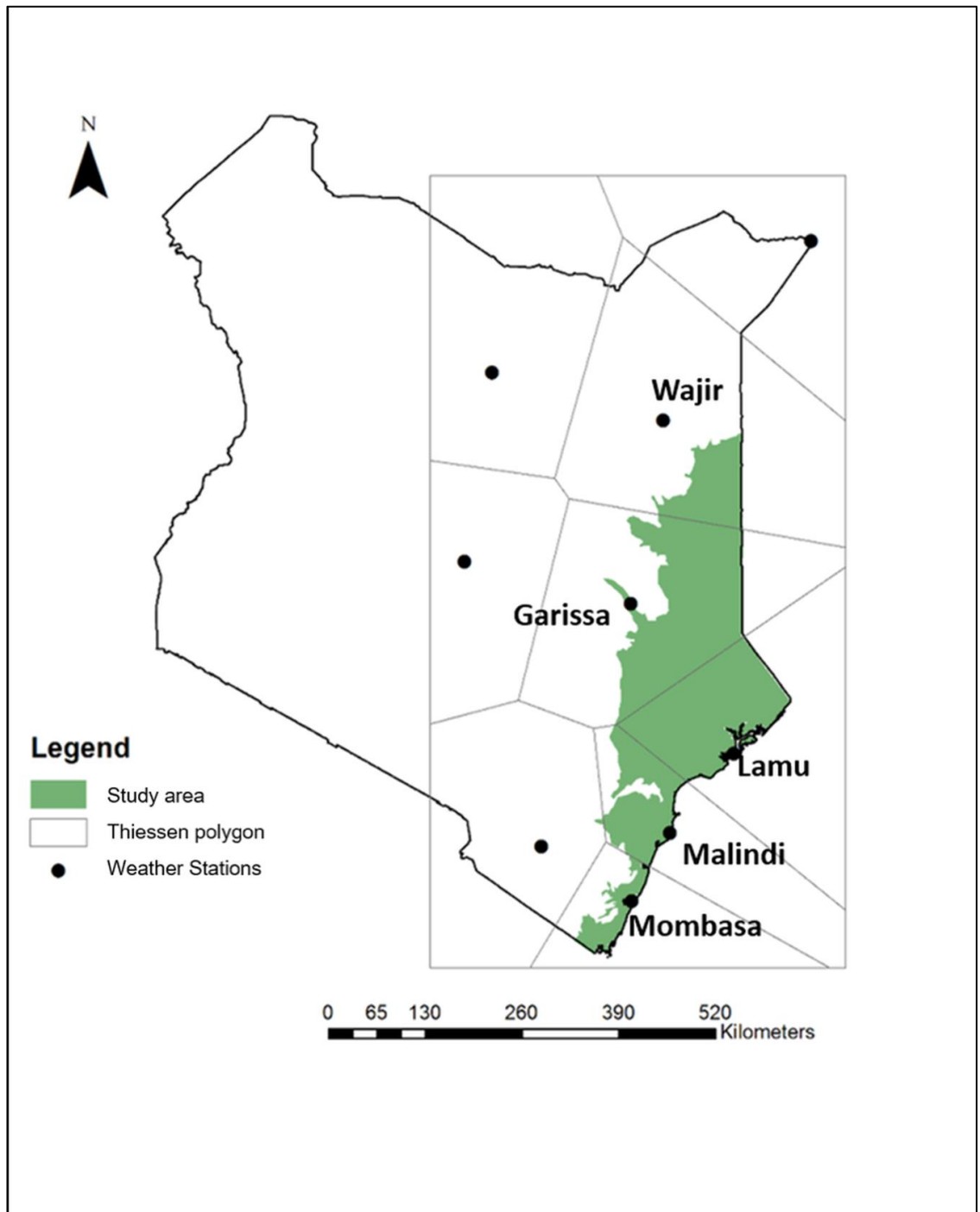

**Figure 2.** Map showing the study area as well as the extent and coverage of each weather station based on Thiessen polygons.

The areas closer to the coast experience relatively lower temperatures than those further away while receiving significantly higher rainfall. The areas further away from the ocean have recorded low rainfall and higher potential evaporation (Table 1). Weather stations with precipitation of below 25 cm/year and potential evaporation above 250 cm/year were considered "dry stations" (Garissa and Wajir) while those stations with precipitation above 90 cm/year and potential evaporation below 200 cm/year were considered "wet stations" (Mombasa, Malindi and Lamu).

**Table 1.** Summary of average annual temperature, precipitation and potential evaporation for each weather station for the 1985–2005 reference period.

| Station | Temperature (°C) | Precipitation (cm/year) | Potential Evaporation (cm/year) |
|---------|------------------|-------------------------|----------------------------------|
| Garissa | 29 | 16.1 | 284 |
| Lamu | 27 | 91.2 | 168 |
| Malindi | 26 | 102.1 | 145 |
| Mombasa | 26 | 102.2 | 171 |
| Wajir | 29 | 8.1 | 287 |

## 2. Materials and Methods

The numerical model used to simulate the water storage in the vadose zone, the cumulative bottom flux (deep percolation to the aquifer) and the surface runoff in this study was HYDRUS-1D, version 4.16.0110. HYDRUS-1D is a one-dimensional variable saturation soil water model that can simulate soil moisture content at any timescale. The model is based on the modified Richards' equation (Equation (1)) using the assumptions that the air phase plays an insignificant role in the liquid flow process and that water flow due to thermal gradients can be neglected [67]:

$$\frac{\partial \theta}{\partial t} = \frac{\partial}{\partial x}\left[K\left(\frac{\partial h}{\partial x} + \cos \alpha\right)\right] - S \tag{1}$$

where $h$ is the water pressure head (L), $\theta$ is the volumetric water content (L$^3$ L$^{-3}$), $t$ is time (T), $x$ is the spatial coordinate (L) (positive upward), $S$ is the sink term (L$^3$ L$^{-3}$ T$^{-1}$), $\alpha$ is the angle between the flow direction and the vertical axis (i.e., $\alpha = 0°$ for vertical flow, 90° for horizontal flow, and 0° < $\alpha$ < 90° for inclined flow) and $K$ is the unsaturated hydraulic conductivity function (L T$^{-1}$) given by:

$$K(h, \, x) = K_s(x)K_r(h, x) \tag{2}$$

where $K_r$ is the relative hydraulic conductivity (-) and $K_s$ the saturated hydraulic conductivity (L T$^{-1}$).

The governing one-dimensional water flow equation for a partially saturated porous medium is described using the modified form of the Richards equation, under the assumptions that the air phase plays an insignificant role in the liquid flow process and that water flow due to thermal gradients can be neglected [68,69].

### 2.1. Model Input

Daily climatic data on a vertical soil profile extending 100 cm downwards from the surface were used as input for HYDRUS-1D. Input requirements for HYDRUS-1D include time information, soil hydraulic properties, initial and boundary conditions and meteorological information (Table 2).

**Table 2.** Input information in the HYDRUS-1D model.

| Input Information | Parameters | Values |
|-------------------|------------|--------|
| Geometry information | Number of layers for mass balance | 1 |
| | Depth of soil profile (cm) | 100 |
| Time information | Time duration (days) | 365 |
| | Time step | $1 \times 10^{-6}$ |
| Water flow-soil hydraulic property model | Model | van Genuchten–Mualem |
| Water flow boundary conditions | Upper boundary condition | Atmospheric BC with surface runoff |
| | Lower boundary condition | Free drainage |

The van Genuchten–Mualem soil hydraulic model was selected in HYDRUS-1D which has a set of soil hydraulic properties (Table 3).

**Table 3.** Soil hydraulic properties for the different soils in the HYDRUS-1D based on the van Genuchten–Mualem model.

|  | Hydraulic Property | Clay | Clay Loam | Sandy Clay Loam | Sand |
|---|---|---|---|---|---|
| $Q_r$ | Residual soil water content, $\theta_r$ (-) | 0.068 | 0.095 | 0.1 | 0.0045 |
| $Q_s$ | Saturated soil water content, $\theta_s$ (-) | 0.38 | 0.41 | 0.39 | 0.43 |
| *Alpha* | Parameter $\alpha$ in the soil water retention function ($L^{-1}$) (1/cm) | 0.008 | 0.019 | 0.059 | 0.145 |
| *n* | Parameter *n* in the soil water retention function (-) | 1.09 | 1.31 | 1.48 | 2.68 |
| $K_s$ | Saturated hydraulic conductivity, $K_s$ ($L\,T^{-1}$) (cm/day) | 4.83 | 6.24 | 31.44 | 712.8 |
| *l* | Tortuosity parameter in the conductivity function (-) | 0.5 | 0.5 | 0.4 | 0.5 |

In the absence of *in situ* data, no simulations were run to test the accuracy of the model. The HYDRUS-1D model was used to simulate infiltration in homogeneous soils, with arbitrary initial water content distributions, subjected to unsteady rainfall, and under the free bottom draining condition to measure deep percolation rate. HYDRUS-1D has been used successfully in numerous studies, providing a good validation of its functionality to model deep percolation [6,70–73]. Ma et al. (2010) [58] further found that the results obtained from the HYDRUS-1D model for cumulative deep percolation were closer to observed results that other models.

Since the flow in the soil profile between the soil surface and the bottom of the model is predominantly vertical in a flat area, the value 1 for vertical was selected for the "Decline in vertical axes". Six observations points were inserted in the soil profile at 0, 20, 40, 60, 80 and 100 cm depths. The soil surface boundary condition was set to atmospheric boundary condition (BC) with surface runoff. In this BC, the potential water flux across the upper boundary is controlled by external conditions. However, the actual flux depends also on the prevailing (transient) soil moisture conditions. The soil surface boundary condition may change from a prescribed flux to a prescribed head type condition and vice-versa. In the absence of surface ponding, the boundary condition is obtained by limiting the absolute value of the flux according to the algorithm explained by Neuman et al. (1975) [74]. Under these conditions, the height of the surface water layer increases due to precipitation and reduces because of infiltration and evaporation. A free drainage condition was selected in the model for the lower boundary where it was assumed that the bottom of the soil column was permeable and water would drain freely towards the underlying aquifer (it was selected as it was assumed that the water table lies far below the domain of interest).

Three different climate scenarios—the reference period, RCPs 2.6 and 8.5—for all five weather stations and for each soil type were run. Input requirements for HYDRUS-1D required surface water fluxes (evaporation and rainfall) as well as soil properties.

## 2.1.1. Soils

Four soils were used in the modelling: the three main types that covered the largest portion of the study area as well as sandy soils for the Lamu area that are typical for the sand dunes overlying coastal aquifers such as the Shela aquifer (an important aquifer for the Lamu island and surrounding areas), as demonstrated by two previous studies [75,76]. The soil classes were determined from data and maps obtained from the Kenya Soil Survey (KSS) database [77]. The four soils occurring in the study area were, (a) clay that covers roughly 50% of the total area, (b) sandy clay loam (20.2 %), (c) clay loam (7.7%) and (d) sand that covers the sand dunes on the Lamu Island. The percentage occurrence of each soil type in the area covered by each weather station is presented in Table 4.

**Table 4.** Percentage occurrence of each soil type in the area covered by each weather station. Figures in bold indicate the main soil in each weather station.

| Soil Type | Occurrence in Area Covered by Weather Station (%) | | | | |
|---|---|---|---|---|---|
| | Garissa | Lamu | Malindi | Mombasa | Wajir |
| Clay | **80.6** | **54.3** | **38.7** | 1.5 | **97.4** |
| Clay loam | 0.1 | 10.0 | 23.1 | 7.9 | 0.0 |
| Sandy clay loam | 4.3 | 33.3 | 9.4 | **28.4** | 1.0 |
| Sand | - | 0.8 | - | - | - |

Sets of soil hydraulic parameters such as soil hydraulic conductivity, infiltration rate, water and holding capacity (pre-set in the model based on van Genuchten–Mualem) were derived from soil types and the pedo-transfer functions in the HYDRUS-1D model based on the United States Department of Agriculture (USDA) database. The soils were classified according to their texture which was based on the relative percentage of silt, clay and sand in each soil as per USDA standards—clay soil contains 40% or more of clay, less than 45% sand and less than 40% silt. Clay loam contains 27–40% clay and 20–40% sand while sandy clay loam contains 20–35% clay, less than 28% silt and at least 45% sand. Lastly, sand contains more than 85% of sand, while the percentage of clay is at 1.5 that of silt with their combined percentage not exceeding 15% [78].

### 2.1.2. Climate Data

Monthly average precipitation, average temperature ($T_{mean}$), maximum temperature ($T_{max}$) and minimum temperature ($T_{min}$) data for the 1986–2005 period were obtained from the Kenya Meteorological Department (KMD). This study focused on the lower and higher RCPs extremes—2.6 and 8.5 for the 2081–2100 timescale to explore the entire range of RCP scenario variations. Projected changes in monthly average ($T_{mean}$), maximum ($T_{max}$) and minimum ($T_{min}$) temperatures and precipitation up to 2100 were derived from the Intergovernmental Panel for Climate Change (IPCC) report on climate change [41] and several studies that had attempted to downscale the projections for Kenya [79,80]. This formed the basis for the calculations of future daily values of both parameters. First, the monthly temperature (T) and precipitation (P) of 1986–2005 were averaged from the observed data provided by KMD. For the RCPs, the average T is projected to increase by an annual average of 1 °C for the RCP 2.6 scenario for the 2081–2100 while the RCP 8.5 average T is projected to increase by an average of 3.7 °C for the same time period. The same projects that P is expected to increase by an average of 2% for the RCP 2.6 scenario and 5% for the RCP 8.5 scenario on average in 2081–2100 period (Table 1). Using these estimates and observed averages, the monthly T and P averages were calculated for one year to represent the average for the 2081–2100 timescale for all 5 weather stations.

As the HYDRUS-1D model needed daily climate data to give a more accurate output, it was required to extrapolate the same from the projected monthly data. The monthly averages for the reference period and both climate change scenarios for the 2081–2100 period were used in a stochastic weather generator (the Weather Generator École de Technologie Supérieure (Wea-GETS), a Weather Generator (WGEN)-like [81]. This is a three-variant (precipitation, maximum and minimum air temperature) single-site stochastic weather generator programmed in Matlab, which is a computer algorithm that uses existing meteorological records to produce a long series of synthetic daily weather data [82,83], to generate average daily data for the reference period and the 2081–2100 period for both RCPs 2.6 and 8.5.

Another important climate parameter needed by the HYDRUS-1D model is the evaporation (the upper boundary condition). In the absence of such data, use of precipitation and temperature data were used to calculate the potential evaporation (evapotranspiration was not used as no vegetation was considered in this study). The Modified Hargreaves method by Droogers and Allen (2002) [84],

which is the most accurate reproduction of the Modified Penman–Monteith approach [85], was used to calculate the daily evaporation, $ET_0$, for the reference period and for both RCP scenarios (Equation (3)).

$$ET_0 = 0.0013 \times 0.408RA \times (T_{mean} + 17) \times (TD - 0.0123P)^{0.76} \qquad (3)$$

where $RA$ is incoming solar radiation, $T_{mean}$ is the mean daily temperature (°C), $TD$ is the daily temperature range ($TD = T_{max} - T_{min}$) and $P$ (cm) is the daily precipitation [86].

Daily precipitation and potential evaporation data for each of the five weather stations was input in the HYDRUS-1D model and run with the corresponding soil types covered by weather station's Thiessen polygon. The data was analyzed and presented first based on each soil type, then grouped in dry and wet stations, and finally summarised using weighted average for each polygon to calculate the water budget of a specified area covered by various soil types.

## 3. Results

### 3.1. Climate Data

The weather stations were divided into two categories: dry and wet stations. The dry stations were Garissa and Wajir that are categorised as Arid and Semi-Arid Lands (ASALs), experienced low annual precipitation (8–16 cm/year) and relatively high average temperature (28.8–28.9 °C) during the reference period. The Mombasa, Malindi and Lamu stations fell under the wet stations with higher precipitation of 91.2–102.2 cm/year and relatively lower average temperatures of 25.8–27.3 °C. In term of individual stations, Mombasa had the lowest average temperature (25.8 °C) and highest rainfall (102.2 cm/year). Malindi has the lowest potential evaporation of 14.5 cm/year while Wajir had the highest of 28.7 cm/year (Figure 3).

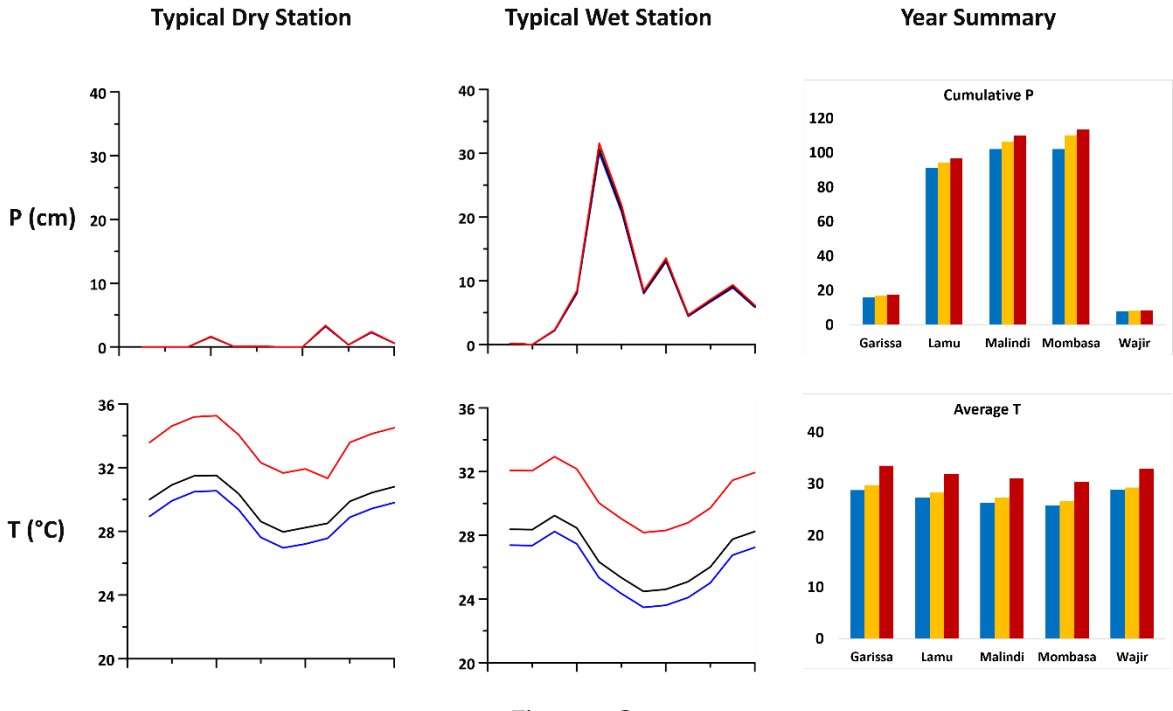

**Figure 3.** *Cont.*

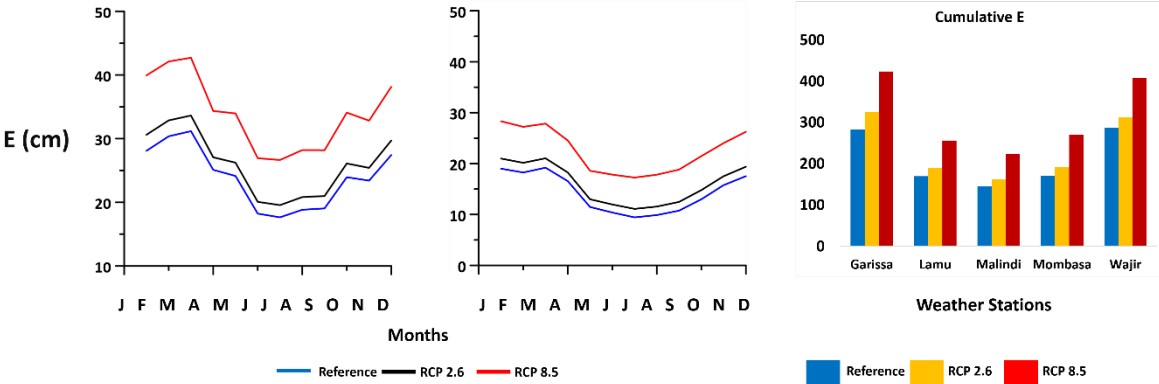

**Figure 3.** Graphical representation of seasonal distribution of precipitation (P), temperature (T) and potential evaporation (E) for typical dry on the left (Wajir) and wet in the centre (Mombasa) stations, as well annual cumulative precipitation, temperature and evaporation for the recorded reference period (1986–2005) and calculated estimates for Relative Concentration Pathways (RCPs) 2.6 and 8.5 (2081–2100) for all weather stations on the right.

Precipitation data results for the reference period as well as for the RCP scenarios for the entire year for all weather stations show high precipitation events in March to May as well as September to December, coinciding with Kenya's rainy seasons. The lowest temperatures of the year coincide with these rainy seasons while the dry season (November to February) experiences the highest temperatures. The rates of high potential evaporation correlate with temperatures i.e., high evaporation occurred when temperatures were high and vice versa (Figure 3).

### 3.2. Simulation Results

The results for different soils types obtained from the HYDRUS-1D model are presented below.

### 3.2.1. Clay Soils

The results show that the average water infiltrated in the study area covered in the clay soils was 35.7 cm/year for the reference period, increasing to 45.9 cm/year for both the RCPs 2.6 and 8.5 scenarios. The average runoff for the same scenarios was 5.5, 6.6 and 6.6 cm/year respectively, while the average water content is 27, 27 and 26% for the respective scenarios. The deep percolation represents ≈11% of the total precipitation, while the surface runoff is 1.7% of the same. The specific scenario results are as follows: the deep percolation for reference period was 3, 24, 79, 70 and 2.3 cm/year for Garissa, Lamu, Malindi, Mombasa and Wajir, respectively. The average water content for the same stations was 27, 30, 30, 31 and 27% of the investigated soil column while cumulative runoff was 9.5, 7 and 11 cm/year for Malindi, Mombasa and Lamu and negligible for Garissa and Wajir (Figure 4).

The deep percolation values increased across all stations for the RCP 2.6 compared to the reference period as highlighted by the results where the cumulative bottom flux was 3, 68, 81, 75 and 2.3 cm/year for Garissa, Lamu, Malindi, Mombasa and Wajir, respectively. The cumulative runoff for Lamu, Malindi and Mombasa is 10.5, 7.5 and 13 cm/year, respectively. The runoff for the dry stations was also negligible while the average water content remained the same as the reference period across all weather stations. For the RCP 8.5 scenario, the deep percolation values increased across all areas from the reference period to levels that were similar to those of the RCP 2.6 scenario as highlighted by the results, i.e., the cumulative bottom flux was 3, 68, 81, 75 and 2.3 cm/year for Garissa, Lamu, Malindi, Mombasa and Wajir, respectively. The cumulative runoff showed an increase compared to both the reference and RCP 2.6 scenarios 11, 8 and 13.9 cm/year for Lamu, Malindi and Mombasa, while those in Garissa and Wajir recorded negligible values. The average soil water content increased to 21, 28, 30 and 30% for Garissa, Lamu, Malindi and Mombasa but remained at 21% for Wajir (Figure 4).

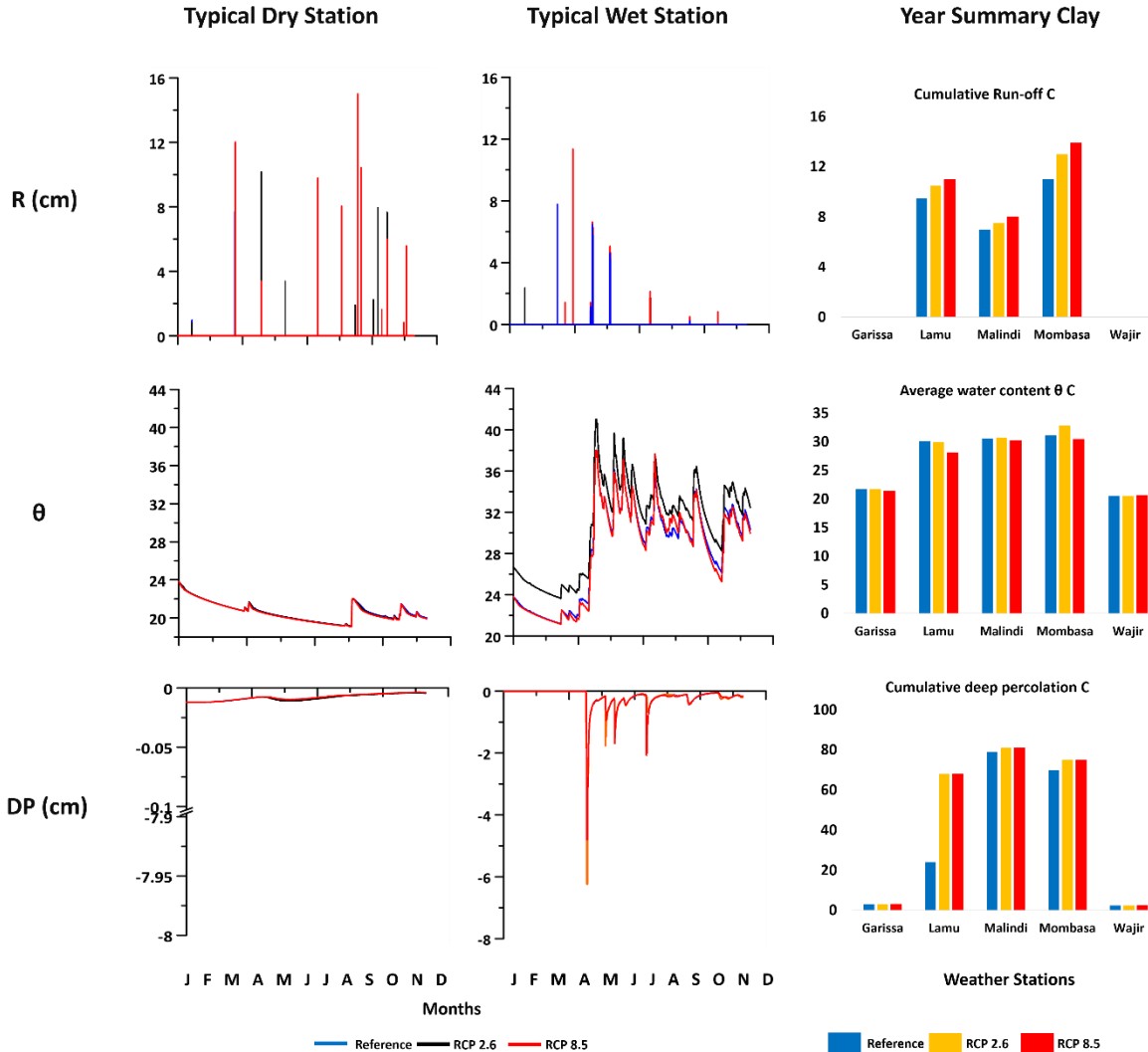

**Figure 4.** Graphical representation of HYDRUS-1D results showing distribution of runoff (R), water content (θ) and deep percolation (D) over a year in clay soils for typical dry station on the left (Wajir) and typical wet station in the centre (Mombasa) stations, as well annual cumulative runoff, average water content and cumulative deep percolation for the recorded reference period (1986–2005), as well as calculated estimates for RCP 2.6 and 8.5 (2081–2100) for all areas on the right.

In all climate scenarios, deep percolation and runoff in the wet stations increased in tandem with increase in precipitation between March and May when it was at its highest level. It reduced after June before increasing again between July and September. A similar trend was noted in the dry stations as well (Figure 4).

### 3.2.2. Clay Loam Soils

The results show that the average water infiltrated across all areas with clay loam soils was 40 cm/year for the reference period. This increased to 49.2 cm/year for the RCP 2.6 scenario and 42.2 cm/year for the RCP 8.5 scenario. The average runoff for the same scenarios was 3.5, 3.6 and 4 cm/year respectively, while the average water content was 29, 29 and 27% for the respective scenarios. The deep percolation represents 12.5% while the surface runoff is 1% of the precipitation. The specific scenario results are as follows: the deep percolation for the reference period was 3.3, 69, 50, 75 and 2.5 cm/year for Garissa, Lamu, Malindi, Mombasa and Wajir, respectively. The average water content for the same stations was 21, 32, 32, 33 and 27% of the investigated soil column while cumulative

runoff was 9.5, 7 and 11 cm/year for Malindi, Mombasa and Lamu and negligible for Garissa and Wajir (Figure 5).

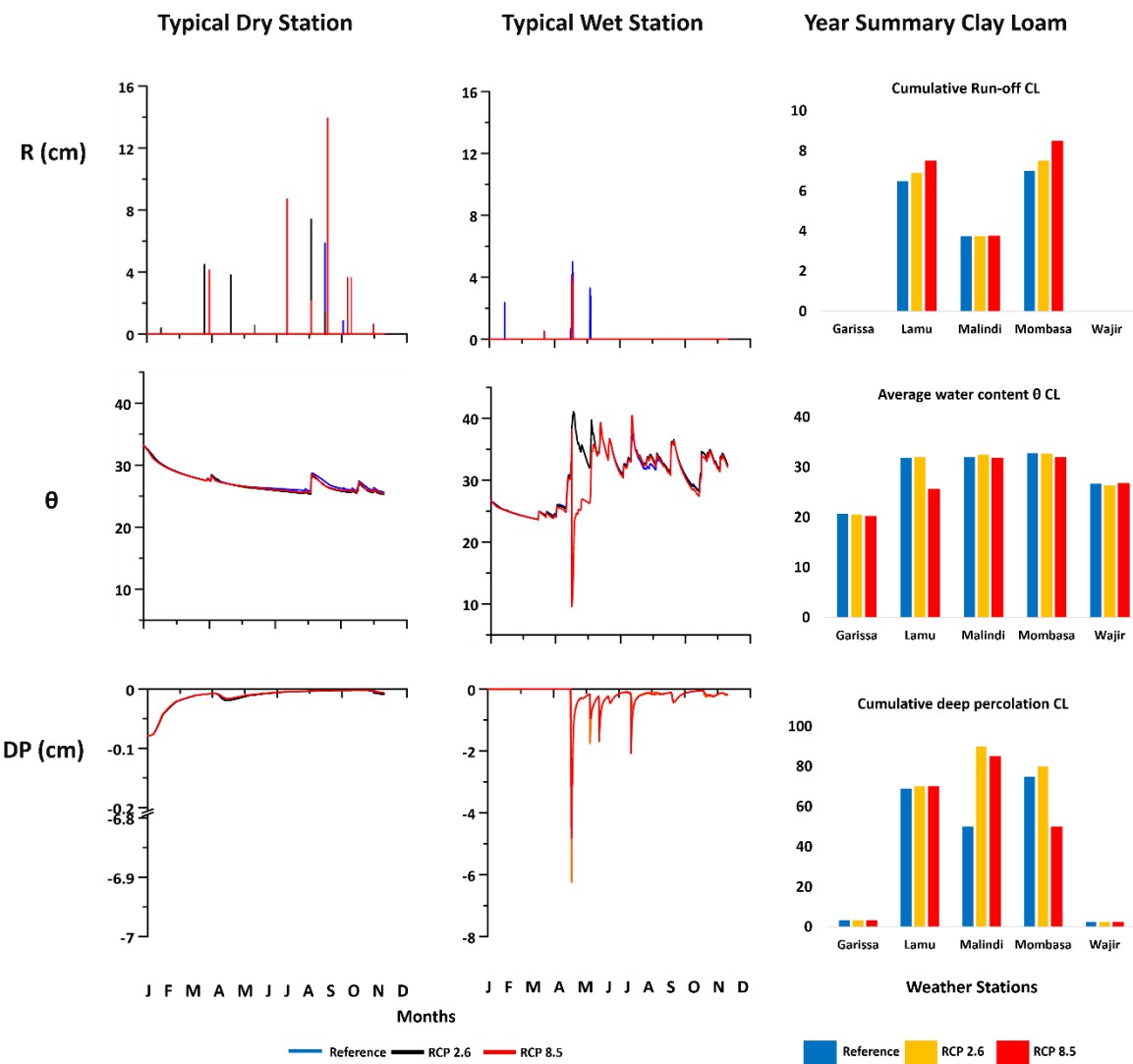

**Figure 5.** Graphical representation of HYDRUS-1D results showing distribution of runoff (R), water content (θ) and deep percolation (DP) over a year in clay loam soils for typical dry station on the left (Wajir) and typical wet station in the centre (Mombasa) stations, as well annual cumulative runoff, average water content and cumulative deep percolation for the recorded reference period (1986–2005), as well as calculated estimates for RCPs 2.6 and 8.5 (2081–2100) for all weather stations on the right.

The deep percolation and runoff values increased across all stations for the RCP 2.6 as highlighted by the results where the cumulative bottom flux was 3.3, 70, 90, 80 and 2.5 cm/year for Garissa, Lamu, Malindi, Mombasa and Wajir respectively, while the cumulative runoff for Lamu, Malindi and Mombasa was 9.5, 7 and 11 cm/year. The runoff for the dry stations was also negligible while the average water content remained the same as the reference period across all areas except the area covered by the Wajir station where it reduced to 26%. For the RCP 8.5 scenario, the deep percolation values increased across all areas from the reference period but were lower than those of the RCP 2.6 scenario as highlighted by the results. where the cumulative bottom flux was 3.3, 70, 85, 50 and 2.5 cm/year for Garissa, Lamu, Malindi, Mombasa and Wajir, respectively. The cumulative runoff showed an increase compared to both the reference and RCP 2.6 scenarios: 7.5, 3.75 and 8.5 cm/year for Lamu, Malindi and Mombasa, respectively, while those in the Garissa and Wajir areas were negligible. The average water content decreased to 20, 26 and 32% of the investigated soil column in Garissa,

Lamu and Mombasa, stayed the same at 32% in Malindi and increased to 27% from the previous two scenarios in the Wajir area (Figure 5).

Just like the clay soils, deep percolation in the wet stations in all climate scenarios increased in tandem with increase in precipitation between March and May when it was at its highest level. It reduced after June before increased again between July and September. The dry stations showed several peaks with the highest increases being observed in April, May, July and September. The runoff in the dry stations showed a similar trend, with the high rates being experienced during the two rainy seasons. However, the highest rates for all climate scenarios were noted in the September/October season. The wet stations showed high runoff rates at the beginning of the year, experiencing the highest rates between March and May with little runoff being recorded during the later seasons of the year (Figure 5).

### 3.2.3. Sandy Clay Loam Soils

The results for the sandy clay loam soils show that the average water recharge across the study area was 52.3 cm/year for the reference period, increasing to 54.1 cm/year for the RCP 2.6 scenario and 54 cm/year for the RCP 8.5 scenario. The average runoff for the same scenarios was negligible while the average water content was 24% of the investigated soil column for all scenarios. The deep percolation represents ≈16% of the precipitation while the surface runoff was negligible. The specific scenario results are as follows: the deep percolation for the reference period was 4.8, 78, 90, 85 and 3.6 cm/year for the Garissa, Lamu, Malindi, Mombasa and Wajir areas, respectively. The average water content for the same stations was 21, 26, 26, 27 and 19%, while the cumulative runoff was negligible for all areas (Figure 6).

The deep percolation and runoff values increased compared to the reference period across all stations for the RCP 2.6 as highlighted by the results where the cumulative bottom flux was 3 4.8, 80, 92, 90 and 3.6 cm/year for Garissa, Lamu, Malindi, Mombasa and Wajir, respectively. The average water content remained the same as the reference period for Lamu, Malindi and Wajir while it increased to 22 and 28% of the investigated soil column for Garissa and Mombasa, respectively. For the RCP 8.5 scenario, the deep percolation values increased across all stations from the reference period but showed a slight decrease compared to the RCP 2.6 scenarios across all stations. Results recorded 4.5, 79, 92, 90 and 2 cm/year for Garissa, Lamu, Malindi, Mombasa and Wajir, respectively, while the cumulative was also negligible for all station. The average soil water content decreased from the RCP 2.5 scenario to 21 and 27% of the investigated soil column in Garissa and Mombasa but stayed the same in the other three stations (Figure 6).

Deep percolation in the wet stations in all climate scenarios increased in tandem with increase in precipitation between March and May when it was at its highest level. This is evident in the RCP 2.6 scenario for the dry stations where the increase was particularly large (from ≈ 0 to 12 cm/year). It reduced after June before increasing again between July and September. A similar trend was noted in the dry stations as well. The runoff in the dry stations showed a similar trend, with the high rates being experienced during the two rainy seasons and the highest rates for all climate scenarios were noted in the September/October season. The wet stations, on the other hand, had near-similar rates throughout the year (Figure 6).

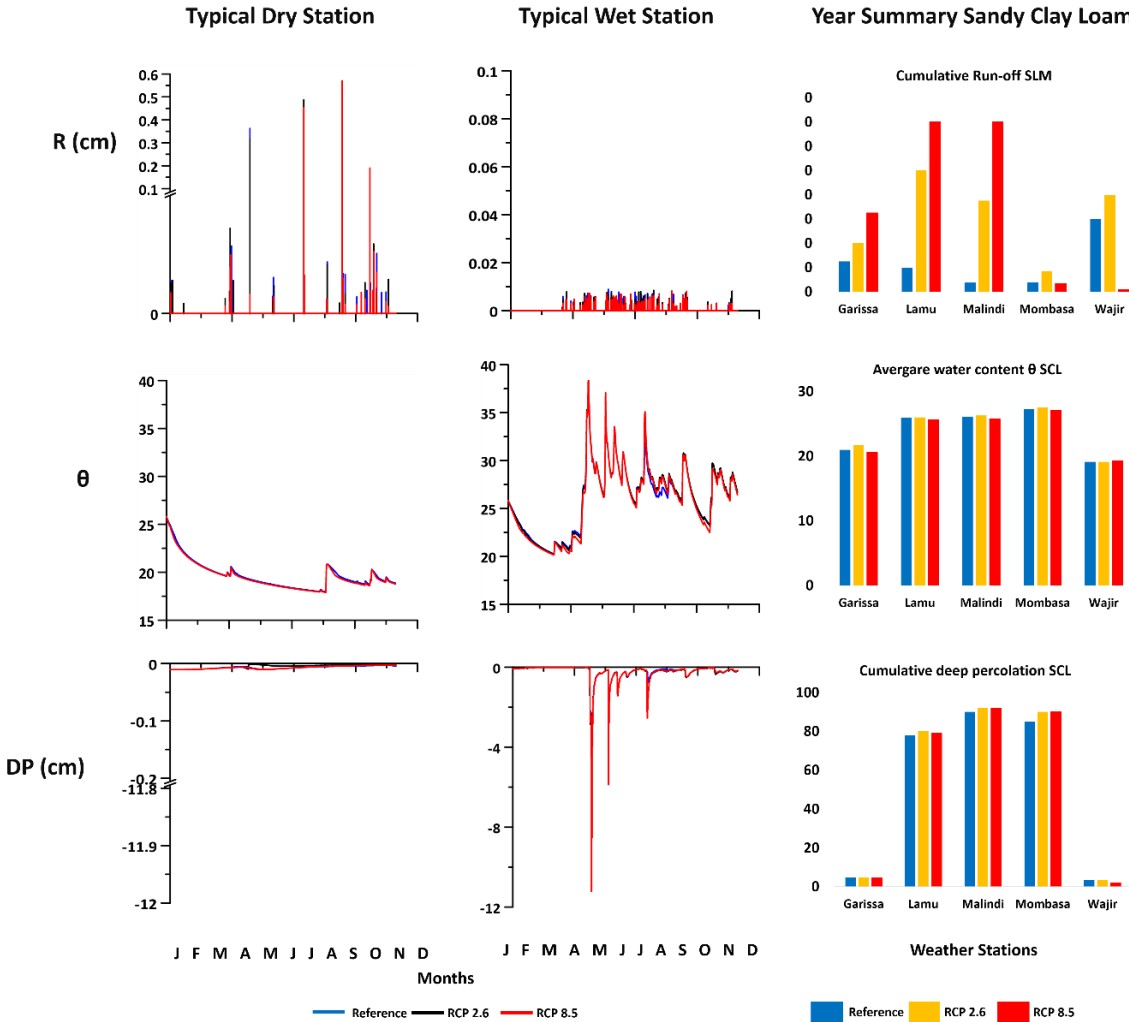

**Figure 6.** Graphical representation of HYDRUS-1D results showing distribution of runoff (R), water content (θ) and deep percolation (DP) over a year in sandy clay loam soils for typical dry station on the left (Wajir) and typical wet station in the centre (Mombasa) stations areas, as well annual cumulative runoff, average water content and cumulative deep percolation for the recorded reference period (1986–2005), as well as calculated estimates for RCPs 2.6 and 8.5 (2081–2100) for all areas on the right.

### 3.2.4. Sandy Soils (Lamu)

For the sand soils in the Lamu area, the water infiltrated to 82 cm/year for the reference period increased to 83 cm/year for RCP 2.6 scenario and further to 85 cm/year for the RCP 8.5 scenario, representing ≈26% of the total precipitation. The runoff was negligible while the average water content was 18, 18.1 and 17.9% of the investigated soil column for the reference period and the RCPs 2.6 and 8.5 scenarios, respectively (Figure 7). Deep percolation also increased in tandem with increase in precipitation between March and May when it was at its highest level. It reduced after June before increasing again in September. The runoff followed a similar trend, with the high rates being experienced during the two rainy seasons. The highest rates for all climate scenarios were noted in March to May during the rainy season (Figure 7). The water content at all depths for the sandy soils in Lamu fluctuated throughout the year following the amount of rainfall precipitated in the area. The average soil water content at all depths for all scenarios was 0.2%.

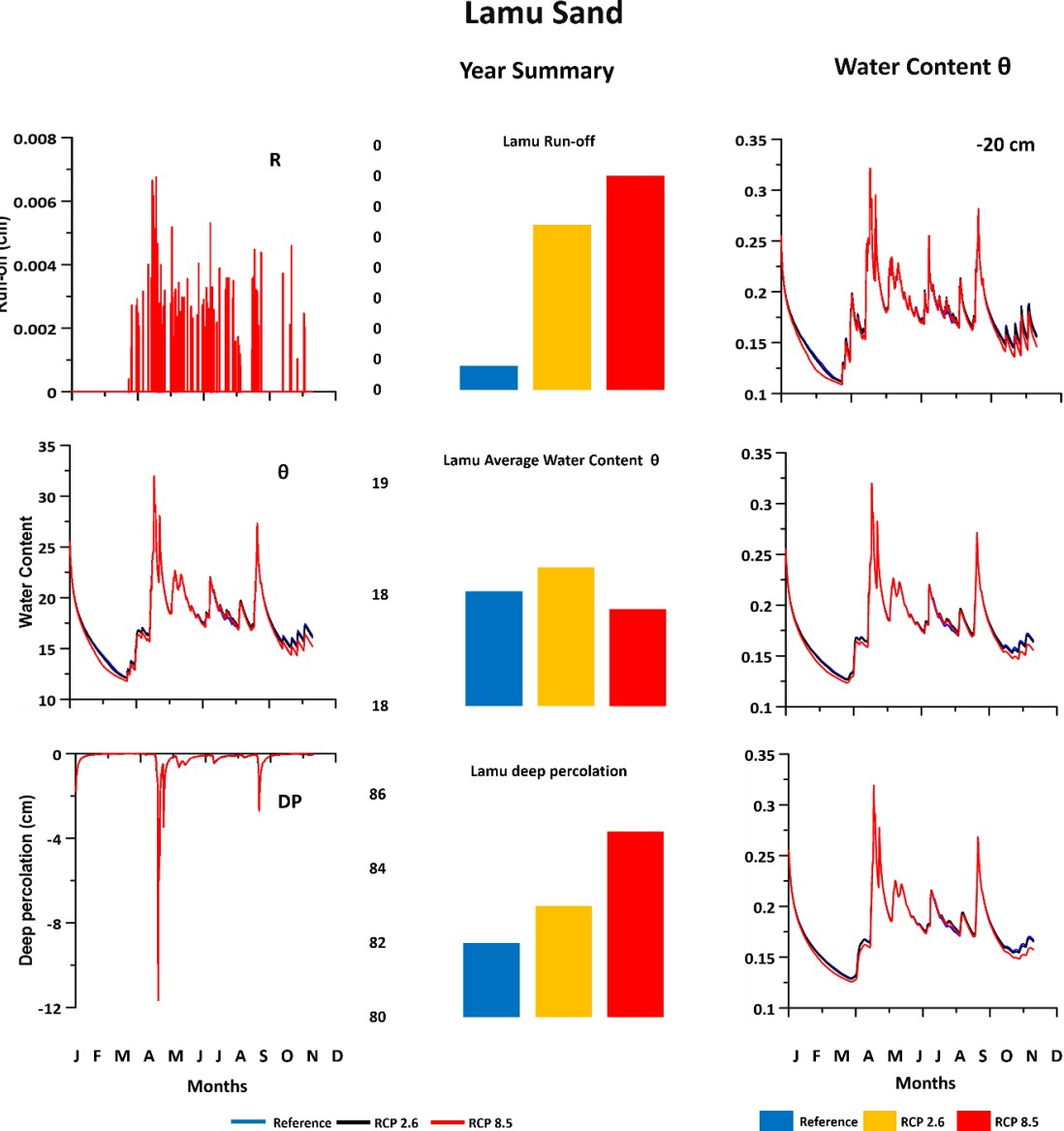

**Figure 7.** Graphical representation of HYDRUS-1D results showing distribution of runoff (R), water content (θ) and deep percolation (DP) in sandy soils of Lamu, as well annual cumulative runoff, average water content and cumulative deep percolation for the recorded reference period (1986–2005), as well as calculated estimates for RCPs 2.6 and 8.5 (2081–2100) for all areas. The graphs on the right represent the water content (θ) at various depths (20, 60 and 80 cm).

The RCPs 2.6 and 8.5 showed larger amounts of deep percolation and surface runoff rates than the reference period. This suggests that the most conservative climate scenario (RCP 2.6), with projected reduction in carbon emissions that keep increase in global temperatures below 2 °C, would still result in more deep percolation of water than the reference period.

### 3.2.5. Water Content at Various Depths in the Soil Column

Generally, the water content for the dry stations at 20 cm depth for all soil types fluctuated throughout the year correlating with the amount of rainfall received in the different areas. The highest water content was recorded in September during second rainy season of the year in all soil types for the reference period and climate scenarios. For the clay and clay loam soils, there was no remarkable difference in soil water content between the reference period and the RCP scenarios. The largest impact on sandy clay loam at this depth was noted in both climate change scenarios, with soil water content

increasing from 22 to 27% for RCPs 2.6 and 8.5 scenarios, respectively. However, at depths of 60 and 80 cm, the water content reduced at a steady rate throughout the year up until September where it increased slightly and reduced again around November. The average water content for all soil types at 20, 60 and 80 cm depths, respectively, for all soil types was found to be 20% (Figures 8 and 9).

The water content for the wet stations at 20 cm, 60 and 80 cm depth for all soil types generally fluctuated throughout the year corresponding with the amount of rainfall received in the areas. All soil types at 20, 60 and 80 cm depths respectively for all soil types was found to be 30%.

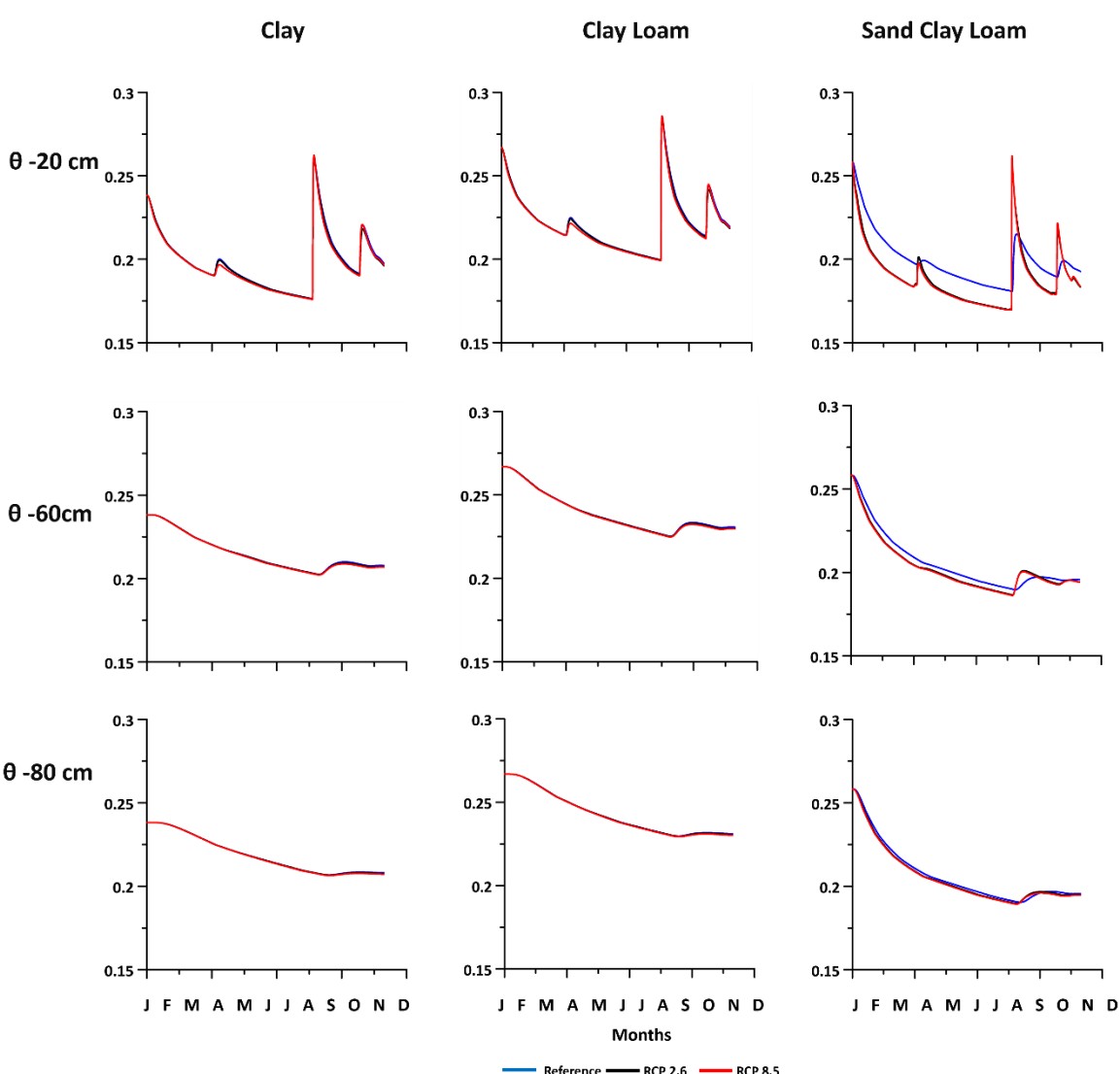

**Figure 8.** Graphs showing the water content (θ) for clay, clay loam and sandy clay loam soils at various depths (20, 60 and 80 cm) of the soil column for a typical dry station (Wajir) for the recorded reference period (1986–2005), as well as calculated estimates for RCPs 2.6 and 8.5 (2081–2100).

## Water content distribution (wet station)

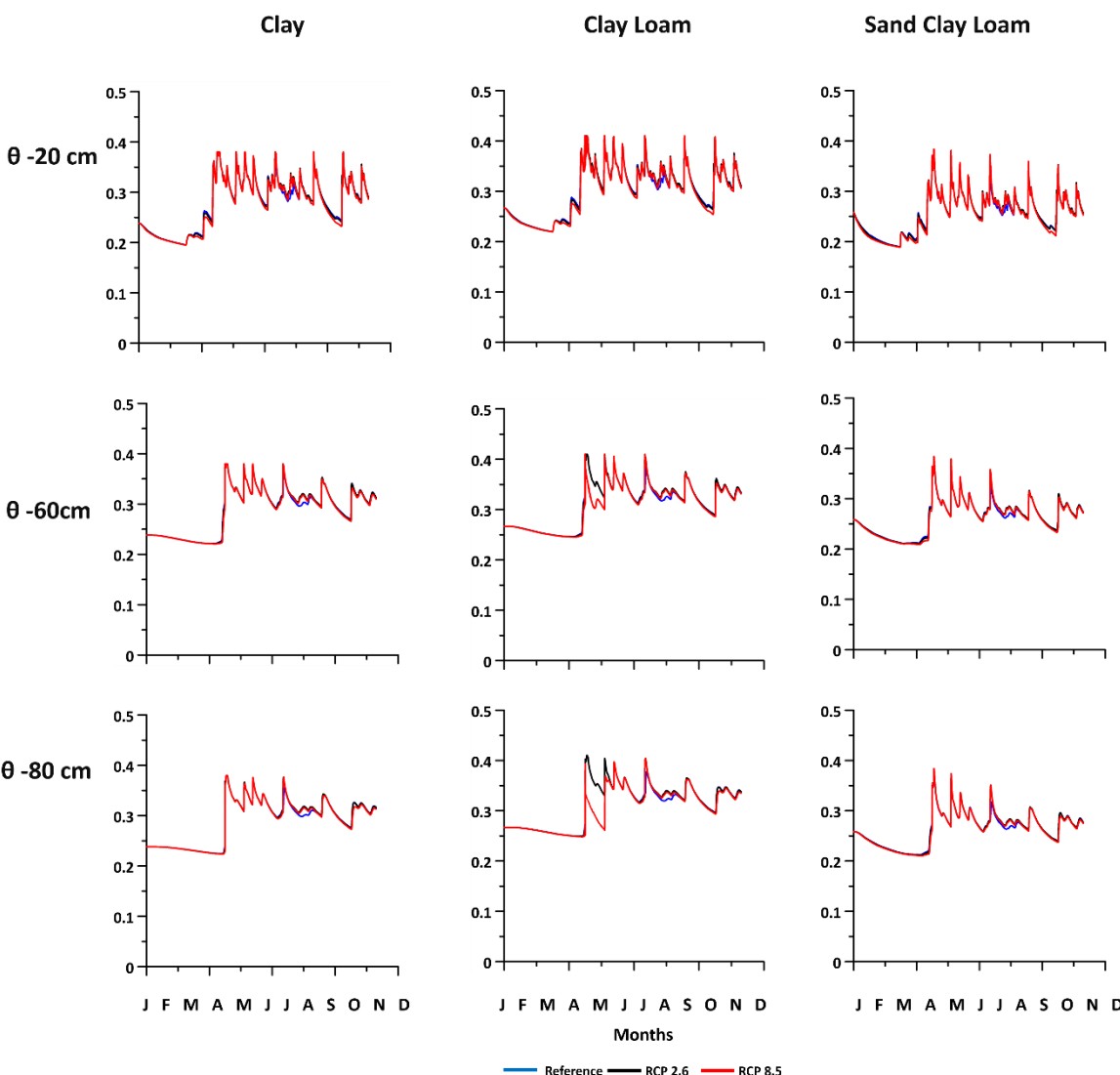

**Figure 9.** Graphs showing the water content (θ) for clay, clay loam and sandy clay loam soils at various depths (20, 60 and 80 cm) of the soil column for typical a wet station (Mombasa) for the recorded reference period (1986–2005), as well as calculated estimates for RCPs 2.6 and 8.5 (2081–2100).

### 3.2.6. Water Budget per Weather Station

A weighted average analysis was carried out to calculate the water budget (deep percolation, runoff and soil water content) for each of the areas influenced by each weather station that was covered by more than one type of soil (Table 5). The results indicate that there was no increase weighted average for DP in the soils covered by the Garissa Thiessen polygon. The RCP 2.6 DP was the same as the reference period (3.31 cm/year) and decreased to 3.25 cm/year in the RCP 8.5 scenario. Garissa recorded no weighted average run off while the water content increased from 21.89 cm/year in the reference period to 21.97 cm/year in the RCP 2.6 scenario. However, it recorded a decrease to 21.58 for the RCP 8.5 scenario. Wajir, the other dry station Thiessen polygon, also showed no change in DP (2.23 cm/year in the reference period and RCP 2.6 scenario) while RCP 8.5 reduced to 2.30 cm/year. The weighted average runoff (7 cm/year) and water content (20.98 cm/year) was constant across all three scenarios. Both Thiessen polygons under the influence of Malindi and Mombasa weather stations had similar trends: both future scenarios recorded increases in DP, with RCP 2.6 having greater increases than

RCP 8.5. Both stations also recorded increases in runoff with RCP 8.5 having greater increases than RCP 2.6. The soil water content in both stations increased in RCP 2.6 while RCP 8.5 had decreases in both stations. For the Thiessen polygon under the influence of last wet station in Lamu, the DP increased from 61.28 cm/year in the reference period to 73.63 cm/year and 73.60 cm/year for RCPs 2.6 and 8.5, respectively. The runoff also increased from 9.66 cm/year (reference period) to 10.6 cm/year and 11.10 cm/year (RCPs 2.6 and 8.5, respectively). However, the weighted average water content reduced from 28.94 cm/year during the reference period to 28.24 cm/year for RCP 2.6 and further to 26.42 cm/year for the RCP 8.5.

**Table 5.** A summary of the weighted averages of the water budget (deep percolation (DP), runoff (R) and soil water storage (water content, SWC)) of all the soil types covered by the 5 different Thiessen polygons for all three scenarios across all weather stations.

| Station/Polygon | Scenario | Weighted Average DP (cm/Year) | Weighted Average R (cm/Year) | Weighted Average SWC (cm/Year) |
|---|---|---|---|---|
| **Garissa** | **Reference** | **3.31** | **0** | **21.89** |
| | RCP 2.6 | 3.31 | 0 | 21.97 |
| | RCP 8.5 | 3.25 | 0 | 21.58 |
| **Lamu** | **Reference** | **61.28** | **9.66** | **28.94** |
| | RCP 2.6 | 73.62 | 10.60 | 28.24 |
| | RCP 8.5 | 73.60 | 11.10 | 26.42 |
| **Malindi** | **Reference** | **74.62** | **6.32** | **30.22** |
| | RCP 2.6 | 85.30 | 6.74 | 30.48 |
| | RCP 8.5 | 83.69 | 7.18 | 29.94 |
| **Mombasa** | **Reference** | **82.58** | **7.94** | **28.63** |
| | RCP 2.6 | 87.56 | 8.89 | 28.96 |
| | RCP 8.5 | 84.29 | 9.80 | 28.29 |
| **Wajir** | **Reference** | **2.32** | **7** | **20.98** |
| | RCP 2.6 | 2.32 | 7 | 20.98 |
| | RCP 8.5 | 2.30 | 7 | 20.98 |

All results above have been summarised, organised and presented in Table 6.

**Table 6.** A summary of the area (in percentage) covered by each soil type in all five weather stations of the study area, the prevailing climatic conditions (precipitation—P and evaporation—E) recorded in each station (reference period 1986–2005), and the expected (positive + or negative −) change for RCPs 2.6 and 8.5 (2081–2100) as well as the soil water budget (deep percolation (DP), runoff (R) and soil water storage (water content, SWC)) for all three scenarios in all soil types across all weather stations. Dashes (-) signify no change in relation to the reference period.

| Station/ Polygon | Soil Type | Area (%) | Scenario | P (cm/Year) | E (cm/Year) | DP (cm/Year) | R (cm/Year) | SWC (%) |
|---|---|---|---|---|---|---|---|---|
| **Garissa** | Clay | 80.63 | **Reference** | **16.1** | **28.4** | **3** | **0** | **22** |
| | | | RCP 2.6 | +0.8 | +4.1 | Ref + 0 | Ref + 0 | Ref + 0 |
| | | | RCP 8.5 | +1.3 | +14 | Ref + 0 | Ref + 0 | Ref − 0.3 |
| | Clay Loam | 0.04 | **Reference** | **16.1** | **28.4** | **3.3** | **0** | **21** |
| | | | RCP 2.6 | +0.8 | +4.1 | - | - | Ref + 0 |
| | | | RCP 8.5 | +1.3 | +14 | Ref + 0.3 | Ref − 1 | Ref − 0.5 |
| | Sandy Clay Loam | 10.58 | **Reference** | **16.1** | **28.4** | **4.8** | **0** | **21** |
| | | | RCP 2.6 | +0.8 | +4.1 | - | - | Ref + 0.7 |
| | | | RCP 8.5 | +1.3 | +14 | Ref − 0.3 | - | Ref − 0.4 |
| **Lamu** | Clay | 54.35 | **Reference** | **91.2** | **17** | **24** | **10** | **30** |
| | | | RCP 2.6 | +2.9 | +1.9 | Ref + 44 | Ref + 1 | Ref − 0.2 |
| | | | RCP 8.5 | +5.6 | +8.6 | Ref + 44 | Ref + 1.5 | Ref − 2 |
| | Clay Loam | 10.00 | **Reference** | **91.2** | **17** | **67** | **7** | **32** |
| | | | RCP 2.6 | +2.9 | +1.9 | - | Ref + 0.4 | Ref + 0.2 |
| | | | RCP 8.5 | +5.6 | +8.6 | Ref + 1 | Ref + 1 | Ref − 6.3 |
| | Sandy Clay Loam | 33.30 | **Reference** | **91.2** | **17** | **78** | **0** | **26** |
| | | | RCP 2.6 | +2.9 | +1.9 | Ref + 2 | - | Ref + 0.1 |
| | | | RCP 8.5 | +5.6 | +8.6 | Ref + 1 | - | Ref − 0.3 |
| | Sand | 0.79 | **Reference** | **91.2** | **17** | **82** | **0** | **18** |
| | | | RCP 2.6 | +2.9 | +1.9 | Ref + 1 | Ref + 0 | Ref + 0 |
| | | | RCP 8.5 | +5.6 | +8.6 | Ref + 3 | Ref + 0 | Ref + 0 |

**Table 6.** *Cont.*

| Station/ Polygon | Soil Type | Area (%) | Scenario | P (cm/Year) | E (cm/Year) | DP (cm/Year) | R (cm/Year) | SWC (%) |
|---|---|---|---|---|---|---|---|---|
| **Malindi** | Clay | 44.72 | **Reference** | **102.1** | **14.5** | **79** | **7** | **30** |
| | | | RCP 2.6 | +4.6 | +1.7 | Ref + 2 | Ref + 0.5 | Ref + 0.2 |
| | | | RCP 8.5 | +7.8 | +7.9 | Ref + 2 | Ref + 1 | Ref − 0.3 |
| | Clay Loam | 23.12 | **Reference** | **102.1** | **14.5** | **50** | **4** | **32** |
| | | | RCP 2.6 | +4.6 | +1.7 | Ref + 40 | - | Ref + 0.4 |
| | | | RCP 8.5 | +7.8 | +7.9 | Ref + 35 | - | Ref − 0.2 |
| | Sandy Clay Loam | 9.39 | **Reference** | **102.1** | **14.5** | **90** | **0** | **26** |
| | | | RCP 2.6 | +4.6 | +1.7 | Ref + 2 | - | Ref + 0.2 |
| | | | RCP 8.5 | +7.8 | +7.9 | Ref + 2 | - | Ref − 0.4 |
| **Mombasa** | Clay | 1.54 | **Reference** | **102.2** | **17.1** | **70** | **11** | **31** |
| | | | RCP 2.6 | +8.1 | +2.1 | Ref + 5 | Ref + 2 | Ref + 1.7 |
| | | | RCP 8.5 | +11.4 | +9.9 | Ref + 5 | Ref + 2.9 | Ref − 0.7 |
| | Clay Loam | 7.89 | **Reference** | **102.2** | **17.1** | **75** | **7** | **33** |
| | | | RCP 2.6 | +8.1 | +2.1 | Ref + 5 | Ref + 0.5 | Ref − 0.1 |
| | | | RCP 8.5 | +11.4 | +9.9 | Ref − 25 | Ref + 1.5 | Ref − 0.9 |
| | Sandy Clay Loam | 28.38 | **Reference** | **102.2** | **17.1** | **85** | **0** | **27** |
| | | | RCP 2.6 | +8.1 | +2.1 | Ref + 5 | - | Ref + 0.4 |
| | | | RCP 8.5 | +11.4 | +9.9 | Ref + 5 | - | Ref − 0.1 |
| **Wajir** | Clay | 97.37 | **Reference** | **8** | **28.7** | **2.3** | **7** | **21** |
| | | | RCP 2.6 | +0.2 | +2.6 | - | - | - |
| | | | RCP 8.5 | +0.5 | +12.1 | - | - | - |
| | Clay Loam | 0.00 | **Reference** | **8** | **28.7** | **2.5** | **0** | **27** |
| | | | RCP 2.6 | +0.2 | +2.6 | - | - | Ref − 0.3 |
| | | | RCP 8.5 | +0.5 | +12.1 | - | - | - |
| | Sandy Clay Loam | 1.02 | **Reference** | **8** | **28.7** | **3.6** | **0** | **19** |
| | | | RCP 2.6 | +0.2 | +2.6 | - | - | - |
| | | | RCP 8.5 | +0.5 | +12.1 | Ref − 1.6 | - | Ref + 0.1 |

### 3.2.7. Climate Change Scenarios

Climate change has been found to impact the water stored in the soils as well as the aquifer systems through changes in recharge, which determine groundwater amount and quality, the surface water and the vadose zone hydrologic balances [40]. Dynamic changes in water contents of the soil profile depend on many natural processes such as rainfall etc., making the climate one of the main drivers in the recharge of aquifers [69]. These effects of climate can be seen from the results in the following ways:

(1) The low rate of deep percolation (<1 cm/year on average) before the rainy season starts in March in all scenarios coincides with the period with little precipitation (less than 1 cm per day) in January and February as recorded across all areas. From Figures 4–7 above, it is clear that the deep percolation increases drastically after March by up to 10 cm/year (from less than 1 cm/year to ≈11 cm/year. At approximately the same time the precipitation increases (Figure 3) in all weather stations (0.2 cm in January to 8.1 cm in March). Moreover, there is a similar trend in surface runoff during the same period of the year. It increases from less than 1 cm/year to as much as 5 cm/year, further demonstrating the influence of climatic conditions on surface runoff. Furthermore, the dry weather stations (i.e., Garissa and Wajir), that had lower rainfall and higher temperatures classified as Arid and Semi-Arid Areas (ASALS) [87], recorded less deep percolation (maximum of 4.8 cm/year) than the wet stations (Lamu, Malindi and Lamu). These ones have higher precipitation and relatively lower mean temperatures and a deep percolation of up to 90 cm/year. This also demonstrates the impact the different seasons of the year have on the soil water budget: The March–May and September–November rainy seasons have the highest runoff and deep percolation, while the drier December–February and June–August register the lower rates. These distinct seasons are important for the deep percolation especially in the ASALs as the rainy seasons coincide with lower temperatures, which reduce the potential evaporation (Figure 3). As a result, deep percolation can occur in these hot, dry areas during the rainy seasons (Figures 4–7) despite there being low precipitation and high potential evaporation. Similar trends were recorded by Sklash et al. (1991) [66] who noted dominant recharge processes for groundwater supplies in the Eastern Kenya were directly as a result of higher rainfall area (>250 mm) and possibly lateral recharge in the lower precipitation area (<250 mm). It should be noted that deep percolation in ASALs also depends on processes that are not included in the present HYDRUS simulations, e.g., occurrence of extreme events such as flooding, droughts, etc. [88].

(2) Future climate change scenarios RCPs 2.6 and 8.5 simulations, which predict an increase in temperature and precipitation (Figure 3, Table 4) [41,79,80], recorded an average high deep percolation and runoff than the reference period. According to the results (Table 4), the average deep percolation is expected to increase by 14% for the RCP 2.6 scenario and by 10% for the RCP 8.5, with the clay, clay loam, sandy clay loam and sandy soils (of Lamu) expected to increase by 29, 23, 3 and 1%, respectively, for the RCP 2.6, and 29, 6, 2 and 4% for the RCP 8.5 scenario. The average runoff is expected to increase by 188% for the RCP 2.6 scenario and by 284% for the RCP 8.5, with the clay, clay loam, sandy clay loam and sandy soils (of Lamu) expected to increase by 13, 5, 159 and 575% respectively for the RCP 2.6, and 20,14, 327 and 775% for the 8.5 scenario. The average soil water content is expected to increase by 1% in the RCP 2.6 scenario and decrease by 2% in the RCP 8.5 scenario. For the different soils, the water content is expected to increase by 1% for clay, sandy clay loam and sand while no change is expected for clay loam soils in the RCP 2.6 scenario. On the other hand, the water content is expected to reduce by 2, 5 and 1% for clay, clay loam and sandy clay loam, with no change expected for sandy soils in the RCP 8.5 scenario.

## 4. Discussion

The type of soil is the one of major factors (among other factors not considered in the study) in controlling soil water retention and infiltration ultimately controlling the amount of water getting into the underlying aquifers through deep percolation [89]. The results of the study further demonstrate the critical role that soil type plays in recharging aquifers. Of the four soil classes studied, clay soils which

cover almost half of the study area (Figure 1), were found to have the lowest deep percolation rate (35.7 cm/year). Conversely, the clay soils had the highest runoff rates of 6 cm/year. The clay loam soils showed the second lowest deep percolation rate (40 cm/year) and second highest runoff (3 cm/year), while sandy clay loam soils had a higher deep percolation rate (52.3 cm/year) and lower runoff rate than the previously mentioned soils (negligible–very low figures that are closer to 0). The sandy soils in the Lamu showed the highest rate of deep percolation of 82 cm/year and the lowest, negligible runoff rate. The soil water content in sandy soils is the lowest average (18 %), followed by sandy clay loam, clay and clay loam (24, 27 and 29 %, respectively).

The differences in deep percolation and runoff are due to the fact that water absorption and retention generally depend on the amount of clay and micro pores present in the soil [78]. Because of their characteristics, clay soils will retain the highest amount of water for longer at the surface therefore increasing the chances of it to pool and be drained on/from the surface or lost through evaporation. As the percentage of clay decreases in clay loam and sandy clay loam, so does the water storage capacity leading to higher deep percolation and lower surface runoff rates as demonstrated by the above results (Figures 4–7 and Table 4). Sandy soils in Lamu (low clay content of less than 15%) showed the highest deep percolation rate (82 cm/year) and lowest runoff rate (negligible), offering further evidence of the role of clay content in soil deep percolation capacities. These results are similar to those in some studies. Elbana et al. (2019) [90], who concluded that higher rates of deep percolation could be found in the loose soil with low slope. Omulabi et al. (2000) [65] studied the influence of soil physical properties on infiltrations rates of sand, clay and silt concluded that infiltration rates varied significantly between the three.

Climate change and soil type may have a specific effect on an area. As highlighted above, results suggest that aquifers underlying soils with higher clay content will have lower recharge rates than those overlain by sandy soils, regardless of the prevailing climate conditions (Table 6). This is significant for the study area, because almost half of it is covered by clay soil. Areas such as those covered by the Garissa and Wajir stations that experience low precipitation and high temperatures (dry stations) are already considered as water scarce areas [91] and almost completely dependent on aquifers, which constitutes the most important source of groundwater in the area [92]. Projected climate change will exacerbate the problem because lower soil water content in areas covered by clay in Table 4 will put the areas at a greater risk of water stress. This is compounded further by the fact that most of the area covered by the Garissa and Wajir stations is made up to 97% by clay soil (Table 3).

However, the similarities between these two future climate change scenarios end there. The RCP 8.5 scenario is expected to have similar or lower deep percolation rates but higher runoff rates than RCP 2.6. This could be explained by the fact that the higher precipitation RCP 8.5 scenario could lead to increased runoff because of soil water saturation due to prolonged rainfall event, short but intense rainfall after long drought and the dry soil have reduced infiltration capacity, among other reasons [93]. Furthermore, the accompanying higher temperatures may result in more evaporation and consequent reduction in water percolating into the soil column as well as the volume being stored. The results in Figure 3 show that the potential evaporation for RCP 8.5 is significantly higher than the other two scenarios. Despite this, climate change is expected to generally have a positive impact on the water resources volumes in the area, as higher deep percolation rates translate to increased volume of water entering the underlying aquifers (increased recharge). This is highlighted in Table 6 where the higher precipitation expected in the two RCP scenarios will contribute to the increased cumulative surface runoff rate as well as the volume of water infiltrating in the vadose zone. These results support the assertion that the change in climate is expected to impact the recharge of aquifers by triggering an increase in infiltration [94]. A similar study carried out by Meixner et al. (2016) [46] on how different aquifers in western USA would respond to climate change showed mixed results—some aquifers showed decline in recharge while others showed a slight increase depending on the projected increase in precipitation. However, it should be noted that increased runoff has negative effects including increased erosion of sandy soils, increased flooding risks especially along the low-lying coastline, deltas, etc. [95].

The section of the study area adjacent to the coastline which is covered by the Mombasa, Lamu and Malindi weather stations is characterised by high precipitation and comparatively lower temperatures, and by having significantly lower clay soil coverage (Table 1) unlike areas covered by Garissa and Wajir stations. The conditions in the Lamu, Mombasa and Malindi areas are promoting infiltration of water and subsequent recharging of aquifers. Areas with sandy soils, like the sand dunes in Lamu, do not face the problems of clay-covered ASALs. High precipitation combined with high infiltration rates will result in relatively higher recharge of the aquifers as quantified by Okello et al. (2015a) [75]. This is an important factor to be considered by water managers as they assess the impact of human interference on the recharge of aquifers along the coastline such as the Shela aquifer that is expected to experience stress due to changing climate and exponential population growth due to land use change Okello et al. (2015b) [76].

Management of groundwater resources in both ASALS and highly populated areas should be a priority for policy makers. They should put special emphasis on measures that increase infiltration especially during the dry seasons and improve water storage during the rainy season. One such measure would be the damming of surface streams to capture the runoff especially in areas with an intermediate clay content where the runoff rate is large. This will reduce the surface runoff giving water time to percolate through the vadose zone [96,97]. Other means of intentional managed aquifer recharge (MAR) suitable for ASALs characterised by hot climates, low rainfall and strong potential evaporation include spreading methods (infiltration ponds/inter-dune ponds—especially along the coastal areas like Lamu), in-channel modifications, open wells, shafts and trenches, borehole recharge and rainwater harvesting, e.g., soakaways from roof-top catchments particularly in highly populated areas [1,98–100]. Another alternative would be to use better agricultural practices like drip irrigation during the dry seasons or planting drought-resistant crops that do not require large volumes of water. Studies like Leterme and Mallants (2011) [33] found that such land use changes and better agricultural practises led to an increase in groundwater recharge.

## 5. Conclusions and Recommendations

This study demonstrated that projected increase in precipitation resulting from climate change would have a positive impact on the recharge of aquifers and soil water storage overall in the study area. The impacts of climate change have a significant, beneficial role in the overall soil water budget. The results show that the infiltration and runoff rates are expected to be higher in 2100 for both RCPs 2.6 and 8.5 scenarios than they were for the reference period. The average deep percolation is expected to increase by 14% for the RCP 2.6 scenario and by 10% for the RCP 8.5 scenario, while the average runoff is expected to increase by 188% for the RCP 2.6 scenario and by 284% for the RCP 8.5 scenario. The average soil water content is expected to increase by 1% for the RCP 2.6 scenario and decrease by 2% for the RCP 8.5 scenario. All this is expected to happen despite projected increase in temperature. The soil properties were also found to have the following effect on the infiltration: clay soils had the lowest rate of infiltration of about 11% of the total precipitation received in the study area and highest surface runoff rate of 1.7% of the total precipitation. Clay loam had the second lowest infiltration rate of 12.5% and a 1.1% surface runoff of the total precipitation received, followed by sandy clay loams with about 16% of the total precipitation percolating to the underlying aquifers. Sandy soils showed the highest infiltration of 26% of the precipitation and recorded the lowest surface runoff of less than 1%, the same as sandy clay loam. The weighted average for each area covered by the Thiessen polygon to show the cumulative water budget for areas covered by more than one soil type showed that deep percolation increased while the weighted average runoff remained constant (dry stations) or increased (wet stations) in both RCP scenarios. However, the weighted average water content increased in the dry stations but showed a decrease in the wet stations in both RCP scenarios.

The methodology used in this study has universal applicability as, with appropriate assumptions, it can be easily adapted to the amount of input data available, to different scales (local to regional) and to different time periods. When modelling infiltration, it is important to consider the impact of human

activities, land use change and vegetation to complement the soil types' role. Also, climate change plays an important role in recharging aquifers, and this is especially important in arid and semi-arid areas such as Garissa and Wajir. Therefore, further studies need to be carried out in the area using more wholesome, robust, multi-disciplinary approaches to include vegetation cover and anthropogenic influences such as agriculture and urbanization, land use and land use change, among other parameters in order to ensure success in tackling water management issues. Management policies that encourage deep percolation, especially in areas covered by clay soil, such as damming of surface runoff and better agricultural practises should be implemented in the Kenyan coastal zone. Uncertainties should also be established and articulated in subsequent studies to improve accuracy of results.

**Author Contributions:** C.O. and M.A. conceived and designed the overall concept and work plan for the research; N.G. and B.M.S.G. consulted on and conceptualised the HYDRUS-1D models alongside M.A.; N.G., along with C.O. and M.A. conceptualised the physical model and helped running the simulations for all three scenarios. C.O., N.G. and M.A. analysed the data; N.W. was the liaison between the first author and the Kenyan authorities and played a critical role in the data collection in Kenya; J.N. supervised revisions and verification of results in the Kenyan context; C.O. wrote the paper with assistance from N.W., J.N. and B.M.S.G. All authors have read and agreed to the published version of the manuscript.

**Funding:** This research was funded by European Commission through the Erasmus Mundus Joint Doctorate program.

**Acknowledgments:** This research was initially carried out as part of the Erasmus Mundus Joint Doctorate Program in Marine and Coastal Management, primarily at the University of Bologna, at the Integrated Geoscience Research Group (I.G.R.G.), which is part of the Interdepartmental Centre for Environmental Sciences Research (CIRSA, Ravenna Campus). Thanks to all our colleagues for the support that they have given us throughout the research period. A special mention is reserved for University of Cadiz for the coordination/administration of the program. A special thanks to Nina Wambiji of the Kenya Marine and Fisheries Research Institute and all staff of Machakos University and Institute of Climate Change and Adaptation (University of Nairobi).

**Conflicts of Interest:** The authors declare no conflict of interest.

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
