# Peer review of "Modelling Projected Changes in Soil Water Budget in Coastal Kenya under Different Long-Term Climate Change Scenarios"

_water, doi:10.3390/w12092455_

Round 1
Reviewer 1 Report
General Comment:
The authors tried to develop numerical models with different soil types to predict soil water budget in coastal Kenya under climate change scenarios. The selected numerical model is HYDRAUS-1D and the considered climate change scenarios are RCP 2.6 and 8.5. The idea of this study is interesting and the purpose is straight forward. However, the applied methods seem logically unsound. The main concern about this study is its experimental design about the model simulations. The setting of the numerical models was unclear and how to calculate the soil-related water budgets was not well explained. I recommend that the manuscript needs a major revision before the publication. When submitting the revised manuscript, the authors should prepare a response to the general comments as well as the following specific comments.
Specific comments:
- The authors developed four numerical models using HYDRAUS-1D to represent four different soil columns, i.e., clay, clay loam, sandy clay loam, and sandy soil. The models were simulated under three difference scenarios of reference, RCP2.6, and RCP8.5.
- It was very unclear that the input data of precipitation, temperature, and evaporation for these models are the same or not. For example, what kind of data were input into the model of clay soil (section 3.2.1) at Wajir (wet) station? How about the other models?
- The soil hydraulic property set in the model should be clear presented, such as a table to list the parameters.
-
- How were the values of soil water budget calculated? For example, surface runoff, and water content, and deep percolation.
-
- In figs. 4 – 7: the soil water budgets at Wajir and Mombasa were shown with different soil types. It is very unclear what the authors try to demonstrate. For example, the calculated surface runoff, water content, and deep percolation of clay soil in fig. 4 represent the typical pattern (distribution) of clay soil at Wajir area? In fig. 5, the calculated soil water budget represent the typical pattern (distribution) of clay loam soil at Wajir area? When comparing figs. 4 and 5, the only difference of input data between these two data set is the soil type. When comparing the dry and wet stations in fig. 4, the only differences to control the water budget are P, T, and E. If clay soil and clay loam soil were covered in Wajir area, what will be its water budget?
- Based on the statement, each 1-D model can only present a point (or a pixel). Then, the authors seem using these simulated results directly for the analysis (only consider the budget caused by the soil which covers the area most). However, each soil type covered certain amount of area in each Thessien polygon (figs. 1 and 2). When analyzing the soil water budget, the effect from each different soil type should be considered. For example, Malindi is covered by clay, clay loam, sandy clay loam, and soil not considered and all of soil water budgets related to different soils should be included in the analysis. A weighted-average method might be applied during the analysis to generate the spatial distribution of soil water budget.
- The physical meaning behind each simulation (figs. 4 – 7 ) was lack and more discussion and explanation should be included in.
- In figs. 4 -7
- The water content is the average of each depth or just a specified depth?
- The scale in each figure should be consistent; otherwise, it is very hard to compare with each other.
- What did the negative sign of DP mean? Downward flux?
- Fig. 7: what is the purpose to show water content at different depths solely in this figure?
- Lines 374 – 377: the statement is inconsistent with fig. 5. The dry station did not show the same trend as the wet station.
- Lines 444 – 449: the statement seems a common sense without the assistance from the simulation models.
- 8 – 9: what is the purpose to show water content at different depths if the soil column was assumed as homogeneous in the vertical direction.
- Table 4: Re-generate the table by present the spatial distribution of water budget in each polygon (station) will delivered more useful information.
Author Response
Please find the responses to the reviewer's comments

Reviewer 2 Report
This article arises a particular interest due to its analysis of the expected water budget in different soils types for some long-term climate change scenarios in a coastal Kenya region with the help of simulations performed by the vertical numerical model HYDRUS.
By the way, generally speaking, I find the results obvious to certain extend. No need to use a numerical model to find out that if the Climate Change Scenarios involves an increase in the Precipitation (and Temperature) the outcome would be an increase in the average runoff, the soil water storage and the percolation, for any type of soil. No analysis considers Evapotranspiration (Real Transpiration) for an increasing Temperature, vegetation cover and/or soil thickness. Anyway, the research uses an acceptable methodology and the quantitative results looks like appropriate, as well as should be expected. In general, the work is well organized and it is worth to be published.
Nevertheless, I found some flaws and inconsistencies in the manuscript that should be addressed in order to reach a better understanding and correctness.
Although the main research methodology is not particularly outstanding (the vertical numerical model HYDRUS mainly based on the Richard`s equation, not developed by the authors) it has been briefly presented in this manuscript (as related to Eq 1 to Eq 5), but unfortunately with some avoidable errors and mistakes. This paragraph should be carefully rewritten or omitted.
There is inconsistencies and misunderstandings in several parts of the manuscript, mainly when describing the results, and in particular as related to the changes in the average runoff, first in the Abstract (lines 23-24), second in 3.2.6 Climate Change Scenarios (lines 518-518), and third in 5. Conclusions and recommendations (lines 613-614): the percentages are not the same (some of them looks erratic) and the results are different, even whether it looks like describing the same results.
In general, the text is well written and organized, but must be checked for errors and innacuracies. I have found some problems along the outline:
RCP 8.5,…..RCP8.6,…are the same? Check along the manuscript
37 line ……contaminants ? …… ….(referred as the
47 line …….herein referred as deep percolation….
77 line ….. experienced ….
107-119 lines RCP 2.6 an 8.5 are not described sufficiently to introduce RCP 4.5 and 6.0 as different, or between these two extremes. Describe better the differences or no mention it.
152 line Rewrite Figure 1 caption.
156-158 lines For a study area where the Potential Evaporation is much higher than
the Precipitation, Real Transpiration should be analyzed.
Figure 2 Check for Thiessen
232 line (RoK, 1982), have neither reference, nor meaning. Check throughout.
259-262 lines Revise paragraph,…… (increase of annual average of 1ºC,… and … increase by 3.7ºC for the same period,…20 years,!..... are out of a comparable scale)
508 line ram….?
627 line …… the soil types role.
650, 652 lines …… thanks to …….
654 line (UoN)….. ?
868 line …..KENYA….. Change format.
Author Response
Please find attached responses to Reviewer 2's comments

Round 2
Reviewer 1 Report
The authors have addressed each comment in detail, except comment#5. The authors only replied that because of length of manuscript, not all of figs. were included in the manuscript. But, the authors still did not explain what will be the water budget at specified area which is covered by more than one kinds of soil types.
In the present form, the manuscript has been improved compared to the previous version. I think the manuscript can be accepted once the above comment is addressed and English language and style are checked.
Author Response
Please find attached the response to the reviewer's comments
